# The Functions and Therapeutic Potential of Heat Shock Proteins in Inflammatory Bowel Disease—An Update

**DOI:** 10.3390/ijms20215331

**Published:** 2019-10-26

**Authors:** Abdullah Hoter, Hassan Y. Naim

**Affiliations:** 1Department of Biochemistry and Chemistry of Nutrition, Faculty of Veterinary Medicine, Cairo University, Giza 12211, Egypt or; 2Department of Physiological Chemistry, University of Veterinary Medicine Hannover, 30559 Hannover, Germany

**Keywords:** heat shock proteins, inflammatory bowel disease, Crohn’s disease, ulcerative colitis, chaperones, therapeutic function

## Abstract

Inflammatory bowel disease (IBD) is a multifactorial human intestinal disease that arises from numerous, yet incompletely defined, factors. Two main forms, Crohn’s disease (CD) and ulcerative colitis (UC), lead to a chronic pathological form. Heat shock proteins (HSPs) are stress-responsive molecules involved in various pathophysiological processes. Several lines of evidence link the expression of HSPs to the development and prognosis of IBD. HSP90, HSP70 and HSP60 have been reported to contribute to IBD in different aspects. Moreover, induction and/or targeted inhibition of specific HSPs have been suggested to ameliorate the disease consequences. In the present review, we shed the light on the role of HSPs in IBD and their targeting to prevent further disease progression.

## 1. Introduction

Inflammatory bowel disease (IBD) is a chronic inflammatory disorder of the gastrointestinal tract that implies dysregulated intestinal homeostasis. Clinically, IBD falls into two major types: Crohn’s disease (CD), which can impact any part of the gastrointestinal tract, and ulcerative colitis (UC), which displays restricted pathology to the colon [1] (see Figure 1). In fact, a complete knowledge of the exact mechanism that leads to the disease is still obscure; however, several factors contributing to the pathogenesis of IBD have been identified [1,2].

IBD results from multifaceted factors which include genetic predisposition, environmental factors, smoking, dietary factors, intestinal microbiota and alterations in the function of the immune system [2,3]. It is believed that a complex interplay between these factors and host innate immunity can initiate the disease [1,3]. In accordance with these observations, our group has recently revealed that pro-inflammatory mediators including TNFα, MCP1 and IL-1β, whose levels are frequently elevated in IBD, can induce endoplasmic reticulum (ER) stress and alter the function and trafficking of key proteins of the intestinal brush border membranes [4]. Owing to the aforementioned complex factors and overlapping mechanisms along with the heterogenic nature of IBD, the disease has been recently identified as an ailment of “complex and variable” nature [5].

Heat shock proteins (HSPs) are highly conserved, stress-induced molecules that are ubiquitously expressed in all eukaryotic cells. These proteins are classically categorized according to their molecular mass into six major families whose members range from 10 to 170 kDa [6,7]. In addition, Kampinga and his colleagues provided an alternative classification that names HSP members in the form of a letter/number combination [8] (Table 1).

Functionally, HSPs differ in their ATP requirements, with high-molecular-weight HSPs needing ATP for their proper function, hence called “ATP-dependent”, and HSPs of small molecular weight (or small HSPs, sHSPs) being “ATP-independent” [9,10]. The principle functions of HSPs are folding of newly synthesized proteins, refolding of misfolded or denatured proteins and prevention of their aberrant aggregation, collectively known as “chaperone activities” [11,12,13]. Other house-keeping functions include protein trafficking and transport between different cellular compartments, besides their secretory and immunological functions [14,15,16,17]. Expression of HSPs is primarily regulated via “heat shock response (HSR)” in which heat shock factor 1 (HSF1) and other transcription factors control the transcription of variant chaperone proteins [18,19]. Notably, HSPs exhibit either a constitutive or an induced expression pattern by which certain members are continuously expressed in non-stress conditions or only expressed/overexpressed after induction by specific stress factors, respectively. Indeed, many physiological and environmental stress factors, such as heat stress, toxic agents, hypoxia, heavy metals, tissue development and differentiation, can induce high expression of HSPs in intestinal epithelial cells [6,20,21] (Figure 2). The upregulation of HSPs accompanies several pathological conditions and inflammatory processes and is involved in the pathogenesis of a multitude of autoimmune and chronic inflammatory diseases such as rheumatoid arthritis [22], atherosclerosis [23], diabetes mellitus [24], myasthenia gravis [25] and IBD [26,27].

It is apparent that the variability and multiplicity of HSPs’ functions remain intriguing and need further in-depth investigations in terms of their cytoprotective, pathogenic or pro-oncogenic criteria [28,29]. Since HSPs are implicated in many house-keeping as well as pathological processes, we aim in the current review to demonstrate and summarize their roles in IBD, particularly those of HSP90, HSP70, HSP60 and small HSPs as major contributors to IBD, and further reveal the potential of their targeting to mitigate the progression or development of the disease.

## 2. Heat Shock Proteins in IBD

### 2.1. HSP90

There is accumulating evidence linking HSP90 expression to the pathogenesis of IBD. Expression of HSP90 was found to be elevated in the intestinal mucosa of patients with UC at the time of diagnosis and to be reduced after therapy [30]. It has been therefore postulated that HSP90, together with other chaperones like HSP70 and HSP60, contributes to the development and maintenance of IBD [30]. Previous contradicting results were reported by Stahl and his colleagues who demonstrated the absence of significant differences between HSP90 levels in healthy and IBD patients [31]. However, fluctuations in the aforementioned levels of HSPs before and after IBD treatments supported the suggestion of utilizing HSPs as useful biomarkers in this disease [30]. A pioneering work by de Zoeten et al. has revealed a key role for Foxp3+ T-regulatory cells (Tregs) in intestinal homeostasis. A reduced Foxp3+ Tregs population has been linked to autoimmune diseases and allograft rejection. These Tregs highly express various histone/protein deacetylases (HDACs) that influence many cellular activities including gene expression, protein function and chromatin remodeling [32]. Interestingly, pan-HDAC inhibition, that has been used for cancer treatment, could enrich Treg production and impact the acetylation status of other non-histone proteins. Of particular importance, targeted inhibition of the HDAC6 isoform (HDAC6i) has been shown to enhance the suppressive functions of Tregs, regulate the HSR and affect HSP90 acetylation [32]. 

Recent studies have shown that mitochondrial HSP90, tumor necrosis factor receptor-associated protein 1 (TRAP1), is highly expressed in UC patients and implicated in UC progression to UC-associated colorectal cancer [33]. On the other hand, expression of the ER chaperone GRP94 has been found lacking in intestinal macrophages (IMACs) of CD patients [34]. Deficient GRP94 in IMACs has been associated with deprivation of tolerance to gut microbiota and further development of chronic inflammation [34,35].

Interestingly, HSP90 inhibitors such as 17-allylaminogeldanamycin (17-AAG) have been found to suppress dextran sulfate sodium (DSS)-induced colitis [36]. In accordance with these results, Novobiocin, another inhibitor of HSP90, has been shown to mitigate DSS-induced colitis and CD45RB^high^ adoptive-transfer colitis in mice via reduction of secretory inflammatory cytokines such as TNF-α [37,38]. In addition, the broad-spectrum antibiotic rifabutin, that has been used in IBD treatment strategies to limit UC symptoms, may also inhibit HSP90 [39]. However, a causal link remains to be proven [40] (see Table 2). Of note, this drug has been used primarily to protect against disseminated *Mycobacterium avium* complex infection in patients with advanced HIV infection, besides its effectiveness in treating multidrug-resistant *Helicobacter pylori* [40,41]. Furthermore, HSP90 inhibition has been actively investigated in many preclinical and clinical studies concerning gastrointestinal and colorectal cancers arising from IBD progression [6,42].

### 2.2. HSP70

Generally, elevated expression of HSP70 has been reported in patients with intestinal inflammatory diseases compared with healthy individuals [21]. Studies with experimentally induced colitis in mice have shown that overexpression of HSPs was not a universal response for all HSPs. In colorectal mucosa, the expression levels of HSP70 and HSP40 were increased, whereas the levels of HSP25, HSP32 and HSP90 remained unaltered [43]. Similar findings, reflecting comparatively high HSP70 mucosal expression, have been reported in UC and CD human patients compared with healthy controls [44]. Notably, HSP70 expression can change in response to treatment type and duration. For instance, a six-month treatment of UC patients with 5-aminosalicylic acid (5-ASA) preparations and probiotics resulted in modulation of the HSP70 expression pattern from high to normal in healthy individuals [30]. These observations support the notion that IBD treatments including chemotherapeutics and antibiotics act to eliminate intestinal commensal bacteria which harbour epitopes necessary for appropriate HSP70 induction [45]. Moreover, the previous results suggest that measurement of HSP levels in the intestinal mucosa imply a strong potential for monitoring the response to treatment in IBD [30].

Previous reports have revealed the significance of HSP70 in the pathogenesis of IBD [21,46]. Ohkawara and co-workers have shown that mice expressing high levels of HSP70 and HSP40, as a result of macrophage migration inhibitory factor gene deletion, could resist pharmacologically induced colorectal inflammation. Subsequent inhibition of HSPs reversed the situation and resulted in the development of colorectal inflammation [43]. Another evidence for the pivotal role of HSP70 in IBD has been speculated from hyperthermia-induced expression of HSPs in mice. In these experiments, mice that had been primarily subjected to heat stress were tolerant to pharmacologically induced colitis compared to non-exposed animals [47].

It is of note that several inflammatory mediators and cytokines, including IL-10, IL-11, IL-1β, TNF-α, can induce HSPs to exert their cytoprotective effect [21]. IL-11 has been demonstrated to confer epithelial-specific cytoprotection via HSP25 but not HSP70 induction in models of intestinal epithelial injury [48,49]. Several in vivo studies point to HSP70 as an anti-inflammatory mediator in intestinal inflammation. In colorectal inflammation, mice highly expressing HSP70 exhibited lower macrophage activity in addition to dropped expression levels of pro-inflammatory cytokines including TNF-α, IL-6, IL-1β compared to control animals. Moreover, the pathologic consequences were milder in animals with augmented HSP70 expression [50]. Furthermore, HSP70 has been reported to trigger IL-10 production with anti-inflammatory effects in model of bacterial infection with *Listeria monocytogenes* and in experimentally induced arthritis [51].

Indeed, besides inflammation-mediated HSP70 induction, many diverse factors including physiological microbiota contribute to HSP70 expression in the intestine. The terminal part of the small intestine, before the colonic junction, has displayed basal expression of HSP70 that has been attributed to potential contact with the colonic bacterial flora, able to impact its expression level in this region [52]. In addition, dietary components, food additives and physical activity such as exercise have a strong impact on HSP70 induction and intestinal epithelial protection [21,53]. All these factors can, therefore, contribute to the clinical course of IBD by HSP-mediated mechanisms. However, these conclusions need further investigation.

Several pieces of evidence implicate HSP70 isoforms in CD pathogenesis. For instance, a single nucleotide polymorphism in HSP70-2 (HSPA1B) has been reported to be involved in a severe clinical course of CD [54,55]. Notably, there are three common genotypes of HSP70-2, AA, AB and BB, resulting from A–G transitions at the 1267 position and producing a *PstI* site. Of those, the heterozygous genotype AB is the most frequently noticed in CD patients, while the AA and BB genotypes are associated with less severe forms of CD [56,57]. In addition, de novo and rare mutations of HSPA1L have been found exclusively in patients with IBD. Further biochemical analyses of these mutations revealed incompetent chaperone activities [58]. Another interesting study has demonstrated that HSPA6 (also known as HSP70B’) can be induced by cigarette smoke, resulting in protection of intestinal epithelial cells via stabilizing anti-apoptotic Bcl-XL [59].

The strong association between HSP70 upregulation in intestinal epithelium and the concurrent protective outcomes against detrimental factors could open new avenues to develop therapeutic approaches for intestinal diseases (see Table 2) [21]. For instance, geranylgeranylacetone, a substance that promotes HSP70 intestinal expression, has been shown to ameliorate or counteract the development of chemically induced colorectal inflammation [60,61]. Similarly, zinc compounds such as polaprezinc exhibited beneficial effects in preventing intestinal inflammation in mice concomitant with overexpression of HSP70 [62]. Polaprezinc has also been shown to protect against acetylsalicylic acid-induced intestinal injury [63]. Furthermore, compounds inhibiting HSP70 expression like quercetin abolished the anti-inflammatory effect of polaprezinc [64]. Interestingly, the anti-IBD drug mesalazine has been shown to augment HSP70 expression in cells of the colorectal mucosa after thermal induction, resulting in a better cytoprotection. Moreover, in an experimental model of large intestine inflammation, high doses of glutamine could upregulate both HSP70 and HSP25 and promote anti-inflammatory activity [65].

On the other hand, special considerations should be taken when using antibiotics for IBD treatment. Oral antibiotics that are used for the treatment of CD have been reported to seriously impact the bacteria within the gut microbiota and consequently disturb HSPs intestinal expression [66,67]. A chronic metronidazole treatment in mice resulted in the reduction of colonic mucosal HSP25 and HSP72 expression and a higher susceptibility to the deleterious effects of bacterial toxins [48]. Similar results of HSP downregulation and even increased mortality were obtained when experimental animals were subjected to combination antibiotic treatments including metronidazole, ampicillin, neomycin and vancomycin [48,68]. Taken together, these observations denote that antibiotics can disrupt the physiological induction of HSPs and, hence, influence the intestinal vulnerability to infection and inflammation [69].

### 2.3. HSP60

Current reports strongly consider the mitochondrial chaperonin HSP60 as a major player in IBD pathogenesis and treatment [78]. To this end, Cappello and his colleagues enumerated several reasons supporting this hypothesis. Among these are: (a) The ability of HSP60 to stimulate pro-inflammatory cytokines [79,80,81,82]; (b) The variability of HSP60 levels in UC mucosa in response to the disease status [73,83]; (c) The implication of HSP60 in other inflammatory conditions such as atherosclerosis [23,84]; and finally (d) The “molecular mimicry” resulting from shared epitopes among human HSP60 and HSP60 from variant intestinal pathogenic microorganisms that ultimately leads to cross-reactivity and development of autoimmunity [25,85,86,87]. In addition, previous studies have revealed that HSP60 is upregulated in the lamina propria of the intestine in IBD patients compared to healthy individuals [83]. These findings suggest a potential role for HSP60 in inflammatory response activation [78]. In accordance with this conclusion, single or co-administration of 5-aminosalicylic acid (5-ASA) with probiotics to IBD patients resulted in mitigation of inflammation and simultaneous reduction of HSP60 expression levels [73,78].

An imbalance in gut microbiota has also been reported to impact HSP60 function in IBD [88]. Dietary probiotics led to lowered levels of HSP60 together with alteration in its post-translational modifications in mice models [89]. These reduced HSP60 concentrations have been attributed to HSP60 secretion into the extracellular milieu either in its soluble free form or via exosomes [90].

Early studies have shown that antibodies against HSP60 could be detected in patients with IBD [91,92]. Moreover, immunohistochemical studies have shown intense staining of HSP60 in mononuclear cells of the intestinal mucosa and submucosa in patients with CD as compared to the mucosal expression in patients with UC [91]. These observations have been evidenced by the co-expression of HSP60 and B7, a marker protein of antigen-presenting cells (APCs) which mediates T-cell activation in patients of IBD. The expression of HSP60 in B7-positive cells has been proposed to play a key role in T-cell activation and stimulation of the inflammatory process [91]. Furthermore, transfer of HSP60-reactive CD8+ T cells into mice resulted in a strong induction of intestinal inflammation that was MHC class I-dependent [93]. Importantly, the transferred HSP60-reactive CD8+ T cells were capable of causing the inflammatory condition regardless of the presence of intestinal bacterial flora. Thus, these findings indicate that HSP60-reactive CD8+ T cells were reactive to cellular HSP60 causing autoimmunity and a severe pathogenic form of colitis [78,93]. Additionally, monoclonal antibodies against HSP60 have been demonstrated to bind HSP60 and reduce inflammatory arthritis and colitis in mice [74]. In addition, studies on diseased children have revealed a strong contribution of HSP60 to the immune and inflammatory responses accompanying paediatric CD [94]. 

It should be noted that several studies link HSP60 to peritumoral inflammation along with clinical course of IBD. Unlike other chaperones, abundant intraepithelial expression of HSP60 has been reported during early stages of colon carcinogenesis, an observation that coincides with that of disturbed homeostasis of intestinal microbiota [95,96,97,98,99]. In cancer cells, HSP60 inhibits apoptosis via pro-caspase 3 binding [100]. Similarly, it associates with cyclophilin D (Cyp-D), preventing its binding to a multi-molecular complex comprising HSP90 and TRAP1, hence hindering tumour cell death [101]. Moreover, HSP60 supports cancer cell survival via a TNF-α-mediated pathway [102]. HSP60 interacts directly with IKKα/β, leading to activation of NF-kB target genes [102]. In addition, secretory forms of HSP60 have been reported to be frequently produced from tumour cells and show distinct posttranslational modifications [90,103,104].

As previously presented, despite the involvement of HSP60 in the molecular mechanisms of diverse diseases including IBD, further investigations are required to fully unravel its molecular roles in IBD. Nevertheless, HSP60 has frequently been suggested as a possible therapeutic target in IBD [78]. In this respect, Meng and his colleagues have summarized the common natural as well as synthetic compounds that can modulate HSP60 levels and suggested that they can be clinically useful for the treatment of diseases including IBD in the near future [105]. 

### 2.4. Small HSPs

The fact that sHSPs are broadly associated with several intestinal pathologies makes them interesting targets to investigate in IBD. Alpha B-crystallin (CRYAB or HSPB5), a significant member of sHSPs, has been recently shown to regulate the intestinal inflammatory response in the intestinal mucosa. In vivo and in vitro studies have implicated CRYAB in the suppression of proinflammatory cytokines (e.g., TNF-α, IL-6, IL-1β and IL-8) through inhibiting the formation of the IKK complex [106]. 

HSP27 is another inducible sHSP whose expression is relatively higher in colonic epithelium as compared to the small intestine. Similar to HSP70, the abundant HSP27 colonic expression has been attributed to stimulation by commensal microbes that exist in large amounts in the colon [48,52]. In addition, many bacteria and bacterial products can trigger HSP27 induction [69]. These include lipopolysaccharide (LPS), short-chain fatty acids (SCFAs), soluble factors from probiotic bacteria including *Lactobacillus GG* and *Bifdobacterium breve* and sporulating factor from *Bacillus subtilis* [76,107,108,109,110]. In line with this information, SCFAs such as butyrate, have been demonstrated to play beneficial roles in the treatment of IBD when supplemented with classical therapeutic agents like mesalazine and corticosteroids [75,76,77].

## 3. Involvement of HSPs in the Progression of IBD to Cancer

It is well known that the functions of HSPs are mainly directed towards cellular protection against various stresses. However, in many occasions, their functional outcomes shift to support tumorigenicity and cancer resistance to chemotherapy [111]. Overexpression of HSPs has been described in many types of cancer including prostate [29], ovarian [112], liver [113], lung [114] and colorectal cancer (CRC) [115,116]. In this respect, CRC stands as a serious, life-threatening complication arising from CD and according to statistics in 2014, it accounts for roughly 15% of IBD-related deaths [69,115]. For instance, HSP27 and HSP70, that are constitutively expressed in the intestinal epithelium, exhibit marked overexpression in CRC [117,118,119]. In addition, HSP27 has been found to support carcinogenesis through anti-apoptotic activities and multidrug resistance [120]. On the other hand, downregulation of specific HSPs has been suggested to contribute to cancer development. Mice lacking HSP70 expression have shown histological features of human IBD-associated colon cancer [121,122]. Moreover, the well-known causative agent of gastric cancer, *H. pylori*, decreases the proliferation of the gastric epithelium via epigenetic mechanisms and downregulation of HSP70 expression [123,124]. These findings indicate that fine-tuning HSPs’ expression is a key process that impacts intestinal homeostasis and regulates HSPs’ proposed functions in either cytoprotection or oncogenesis [69]. For this reason, there have been several endeavours to inhibit HSP-induced carcinogenesis and tumorigenesis through the use of HSP inhibitors or mutant HSPs. These efforts are aimed at sensitizing human colorectal cancer cells to chemotherapy or radiation therapy [117,125,126]. Furthermore, microbe-derived HSP vaccines against *H. pylori* have been developed and could induce protective immunity concurrent with a diminished inflammatory response [127].

## 4. Conclusions and Perspectives

Undoubtedly, HSPs are involved in intestinal homeostasis and pathophysiology. Many stress factors causing IBD can trigger the expression of HSPs. HSP90, HSP70, HSP60, HSP27 are key chaperones that protect against damage of the colorectal mucosa, and their increased expression affords cytoprotection and restricts IBD progression. In contrast, however, HSPs supports disease progression in the intestinal mucosa, according to stage of the disease and other poorly understood mechanisms. Therefore, understanding the regulation of HSP expression is essential to help slow down the clinical course of the disease. Taken together, pharmacological modulation of HSPs’ expression represents a significant therapeutic approach, which may potentiate the current treatment regimens in patients with IBD. 

## Figures and Tables

**Figure 1 ijms-20-05331-f001:**
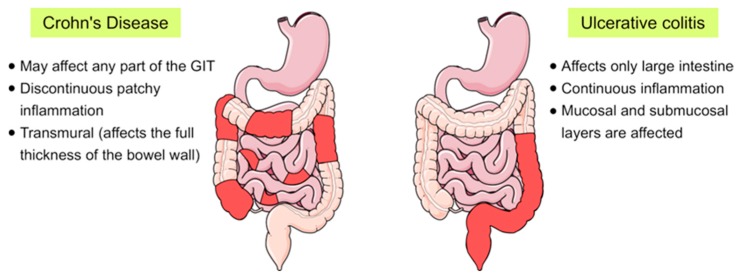
Schematic representation showing the main differences between the two main forms of inflammatory bowel disease (IBD), Crohn’s disease and ulcerative colitis, in terms of their location and the pattern of the affected areas (red-coloured) in the gastrointestinal tract (GIT).

**Figure 2 ijms-20-05331-f002:**
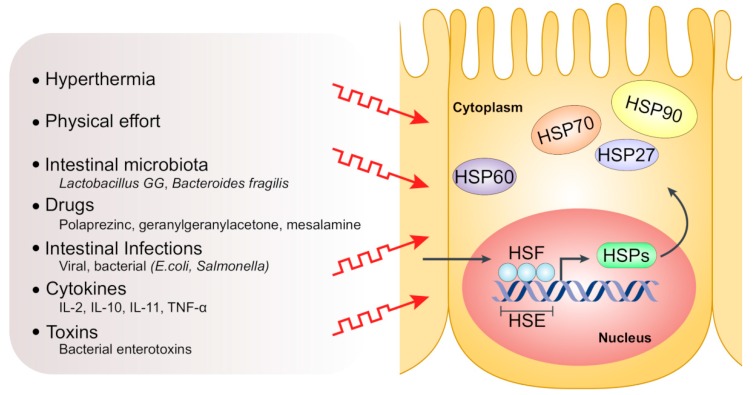
Factors inducing HSP expression in intestinal epithelial cells. In response to various stresses such as hyperthermia, intestinal infections, toxins and drugs, activated trimeric heat shock factor (HSF) translocates to the nucleus and binds to the heat shock element (HSE) in the promotor region of *HSP* genes, thus triggering HSP transcription.

**Table 1 ijms-20-05331-t001:** Summary of heat shock proteins (HSP) families and their common members.

HSP Family	Alternative Family Name	Number of Members	Common Selected Members
HSP110	HSPH	4	HSPH1 (HSP105), HSPH2 (HSP110, HSPA4)
HSP90	HSPC	5	HSPC2 (HSP90α), HSPC3 (HSP90β), HSPC4 (GRP94, HSP90B1, GP96, endoplasmin), HSPC5 (TRAP1, HSP75, HSP90L)
HSP70	HSPA	13	HSPA1A (HSP70-1), HSPA1B (HSP70-2) HSPA5 (BIP, GRP78), HSPA6 (HSP70B′), HSPA8 (HSC70), HSPA9 (GRP75)
HSP60 and HSP10 (Chaperonins)	HSPD and HSPE	14	HSPD1 (HSP60), HSPE1 (HSP10)
HSP40	DNAJ	50	DNAJA1, DNAJB1 (HSPF1 and HSP40), DNAJC1
Small HSPs	HSPB	11	HSPB1 (HSP27), HSPB4 (CRYAA) and HSPB5 (CRYAB)

**Table 2 ijms-20-05331-t002:** Modulators of HSPs as a therapeutic strategy in preclinical and clinical studies of IBD.

HSP Member	Compound	Action	Reference
HSP90	17-Allylaminogeldanamycin (17-AAG)	- N-terminal ATPase-targeted HSP90 inhibitor - Inhibits dextran sulfate sodium (DSS)-induced colitis - Increases the production of anti-inflammatory cytokines including interleukin (IL)-10 in the colon - Increases the suppressive function of Foxp3+ Tregs in vitro and in vivo.	[36,38]
	Novobiocin	- Inhibits the HSP90 C-terminal ATPase - Attenuates dextran sulfate sodium-induced colitis and CD45RB^high^ adoptive-transfer colitis	[37]
	Rifabutin	- Inhibits HSP90 - Treats multidrug-resistant *Helicobacter pylori*	[40,41]
HSP70	Geranylgeranylacetone	- Enhances HSP70 expression - Acyclic polyisoprenoid that protects the stomach from mucosal injury - Protects against oxidative stressors including monocrolamine (NH2Cl) and 2,4,6- trinitrobenzene sulfonic acid (TNBS) in mice.	[60,61]
	Polaprezinc (*N*-(3-aminopropionyl)-l-histidinato zinc)	- Increases the expression of HSP70 - Anti-inflammatory and anti-ulcer drug - Protects against acetylsalicylic acid-induced intestinal injury as well as DSS-induced colitis in mice	[62,63]
	Mesalamine	- Supports thermal induction of HSP72 in intestinal epithelial cells - Supports intestinal mucosal integrity and reduces inflammatory response	[70,71]
	Glutamine	- Increases the expression levels of both HSP70 and HSP27 in intestinal cells - Protects intestinal cells against inflammation-induced stress - Pharmacologic doses of glutamine lessen DSS-induced colitis in vivo	[65,72]
HSP60	5-Aminosalicylic acid (5-ASA)	- Downregulation of HSP60 together with reduction of inflammation	[73]
	Prozumab	- Humanized anti-HSP monoclonal antibody able to bind HSP60 - Counteracts murine inflammatory arthritis and colitis - Stimulates interleukin 10 (IL-10) secretion from naive human peripheral blood mononuclear cells (PBMCs) and decreases release of IFN-γ and IL-6 from anti-CD3-activated human PBMCs	[74]
HSP27	Butyrate	- Enhances the expression of HSP27 - Supports intestinal epithelial cells function and integrity - Oral butyrate may augment the potency of oral mesalazine in active ulcerative colitis (UC)	[75,76,77]

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
