# Peer review of "The Functions and Therapeutic Potential of Heat Shock Proteins in Inflammatory Bowel Disease—An Update"

_ijms, 2019, doi:10.3390/ijms20215331_

Round 1

Reviewer 1 Report

In general, the manuscript is well-written. In this review, the authors have summarized the role of HSPs in IBD and their targeting to prevent further disease progression. Few minor points are listed below:

Please use only the abbreviated form in subsequent sentences. There are many places all over the manuscript, e.g., page 7 line 206, 207, 219 (CD/UC). Also, on page 7 line 225 TRAP1. In page 4 line 106, please write Mycobacterium avium complex infection, not disease. In figure 1, features for CD should be transmural instead of “mainly submucosal layer”. In figure 1, features for UC should be Mucosal and submucosal instead of mucosal layer only. Please use Rapid Review Pathology 4th Edition page 457 for the correction and additional information. In figure 2, please use some examples with parentheses for each known factors. E.g., drugs (….), cytokines (…).

Author Response

Reviewer # 1

We thank the reviewer for the favorable comments and constructive suggestions, which we have addressed in the revised version of the manuscript. In what follows are our responses to the specific points raised in the review.

In general, the manuscript is well-written. In this review, the authors have summarized the role of HSPs in IBD and their targeting to prevent further disease progression. Few minor points are listed below:

Please use only the abbreviated form in subsequent sentences. There are many places all over the manuscript, e.g., page 7 line 206, 207, 219 (CD/UC). Also, on page 7 line 225 TRAP1. In page 4 line 106, please write Mycobacterium avium complex infection, not disease.

Response:

The abbreviated forms are used all over the revised manuscript as suggested by the reviewer. In addition, “Mycobacterium avium complex disease” was corrected into “Mycobacterium avium complex infection”.

In figure 1, features for CD should be transmural instead of “mainly submucosal layer”.

In figure 1, features for UC should be Mucosal and submucosal instead of mucosal layer only. Please use Rapid Review Pathology 4th Edition page 457 for the correction and additional information.

In figure 2, please use some examples with parentheses for each known factors. E.g., drugs (….), cytokines (…).

Response:

We edited Figure 1 and Figure 2 according to the reviewer’s comments and recommendations.

Reviewer 2 Report

A very interesting review oh HSP role in IDB that could be published with small correction (not in the substance of the article)

Just small editing corrections:

do not use the abbreviation in the title arrange table 2 - I would not list the HPS as a column, but in a raw and then the medication and effects. I would not use the centered function in column 3 no comma in line 51

Author Response

Reviewer # 2

We thank the reviewer for the favorable comments and constructive suggestions, which we have addressed in the revised version of the manuscript. In what follows are our responses to the specific points raised in the review.

A very interesting review oh HSP role in IDB that could be published with small correction (not in the substance of the article)

Just small editing corrections:

Do not use the abbreviation in the title

Arrange table 2

I would not list the HPS as a column, but in a raw and then the medication and effects. I would not use the centered function in column 3 no comma in line 51.

Response:

We changed the review title to include the "Heat Shock Proteins” rather than the abbreviation "HSPs". We also edited the format of the table according to the journal requirements.

Reviewer 3 Report

The review by Hoter et al., entitled “The functions and therapeutic potential of HSPs in inflammatory bowel disease-an update” is well written, well-constructed and of great interest in the field of IBD.

I suggest some minor modifications to the manuscript.

Line 38: “the pro-inflammatory mediators including TNF-alpha, MCP1, IL1-beta that have been observed in IBD”. This sentence is not clear. What do the authors mean by “observed in IBD”? Please be more precise.

line 52: “protein folding of newly synthetized proteins”. Remove the first “protein”.

Figure 2: “Intestinal microbiota” instead of “intestinal flora”.

Figure 2 legend: “In response to variant stress”. What does “variant” mean in this context?

lines 78-80: “HSP90 expression elevated in UC patients”. Please precise in which tissue? Blood cells, Intestinal mucosa? IECs? Sub-mucosa?

lines 104-105: “Rifabutin has been found to treat IBD via HSP90 inhibition”. This sentence should be softened. The antibiotic may inhibit HSP90 (please add a reference) and limit UC symptoms (please add a reference). However, the causal link remains to be proven (please add a reference showing this link if previously described).

line 106: “Mycobacterium avium” should be in italic

line 109: Please give more details about the preclinical and clinical studies cited and the results obtained.

line 127: “could resist TO pharmacologically-induced colorectal inflammation”

line 142: Listeria monocytogenes should be in italic

line 175: “antibiotics used as immunosuppressive therapies for treatment of CD”. Antibiotics are not immunosuppressive molecules. Please rewrite this sentence.

line 192: what does “variant” mean in this context?

line 194: “Lamina propria of intestinal epithelial cells” What does that mean? It is either lamina propria OR IECs… Lamina propria shoud be in italic

A table in describing the role of each HSP in IBD would be useful to summarize the review. The table could detail what is known for each HSP in IBD context, the models used in the diverse studies cited…

Overall, this review perfectly summarize the current knowledge about HSPs in IBD and is of great interest.

Author Response

Reviewer # 3

We thank the reviewer for the favorable comments and constructive suggestions, which we have addressed in the revised version of the manuscript. In what follows are our responses to the specific points raised in the review.

The review by Hoter et al., entitled “The functions and therapeutic potential of HSPs in inflammatory bowel disease-an update” is well written, well-constructed and of great interest in the field of IBD.

I suggest some minor modifications to the manuscript.

Line 38: “the pro-inflammatory mediators including TNF-alpha, MCP1, IL1-beta that have been observed in IBD”. This sentence is not clear. What do the authors mean by “observed in IBD”? Please be more precise.

Response:

We changed the sentence “pro-inflammatory mediators including TNFα, MCP1 and IL-1β that have been observed in IBD, can induce ER stress and alter the function and trafficking of key proteins of the intestinal brush border membranes” to read “In accordance with these observations, our group has recently revealed that pro-inflammatory mediators including TNFα, MCP1 and IL-1β, whose levels are frequently elevated in IBD, can induce ER stress and alter the function and trafficking of key proteins of the intestinal brush border membranes”.

line 52: “protein folding of newly synthetized proteins”. Remove the first “protein”.

Figure 2: “Intestinal microbiota” instead of “intestinal flora”.

Response:

The corrections were made as recommended by the reviewer.

Figure 2 legend: “In response to variant stress”. What does “variant” mean in this context?

Response:

We edited Figure 2 to be more detailed and expanded the legend to be more informative.

lines 78-80: “HSP90 expression elevated in UC patients”. Please precise in which tissue? Blood cells, Intestinal mucosa? IECs? Sub-mucosa?

Response:

We amended the sentence to include the specific tissue (intestinal mucosa) that highly expresses HSP90.

lines 104-105: “Rifabutin has been found to treat IBD via HSP90 inhibition”. This sentence should be softened. The antibiotic may inhibit HSP90 (please add a reference) and limit UC symptoms (please add a reference). However, the causal link remains to be proven (please add a reference showing this link if previously described).

Response:

We changed the sentence accordingly.  

line 106: “Mycobacterium avium” should be in italic

line 142: Listeria monocytogenes should be in italic

Response:

The edits were made as suggested.

line 109: Please give more details about the preclinical and clinical studies cited and the results obtained.

Response:

The cited references in this respect are comprehensive reviews on targeting HSP90 in many cancers including gastrointestinal cancer which may arise from IBD progression. In this context, we think that the sentence is clear and draws the reader's attention for further reading in this direction.

line 127: “could resist TO pharmacologically-induced colorectal inflammation”

Response:

We believe the sentence is grammatically correct in its form.

line 175: “antibiotics used as immunosuppressive therapies for treatment of CD”. Antibiotics are not immunosuppressive molecules. Please rewrite this sentence.

Response:

The sentence was corrected.

line 192: what does “variant” mean in this context?

line 194: “Lamina propria of intestinal epithelial cells” What does that mean? It is either lamina propria OR IECs… Lamina propria shoud be in italic

Response:

The sentences were corrected and explained according to the reviewer’s recommendation.

A table in describing the role of each HSP in IBD would be useful to summarize the review. The table could detail what is known for each HSP in IBD context, the models used in the diverse studies cited…

Overall, this review perfectly summarize the current knowledge about HSPs in IBD and is of great interest.

Response:

A Table that summarizes the review would be indeed useful depending on the length and the body of information included. The current mini-review, however, compiles in a summarized, easily legible and digestible manner the knowledge on HSPs and their potential implication in IBD and we think that an additional Table would not add much to this mini-review.
